# Unraveling Synergism between Various GH Family Xylanases and Debranching Enzymes during Hetero-Xylan Degradation

**DOI:** 10.3390/molecules26226770

**Published:** 2021-11-09

**Authors:** Samkelo Malgas, Mpho S. Mafa, Brian N. Mathibe, Brett I. Pletschke

**Affiliations:** 1Enzyme Science Programme (ESP), Department of Biochemistry and Microbiology, Rhodes University, Grahamstown 6140, South Africa; brianmathibe7@hotmail.com (B.N.M.); b.pletschke@ru.ac.za (B.I.P.); 2Department of Biochemistry, Genetics and Microbiology, University of Pretoria, Pretoria 0028, South Africa; 3Carbohydrates and Enzymology Laboratory (CHEM-LAB), Department of Plant Sciences, University of the Free State, Bloemfontein 9300, South Africa

**Keywords:** α-l-arabinofuranosidase, α-d-glucuronidase, β-xylanase, glycoside hydrolase, hetero-synergy, xylan degradation

## Abstract

Enzymes classified with the same Enzyme Commission (EC) that are allotted in different glycoside hydrolase (GH) families can display different mechanisms of action and substrate specificities. Therefore, the combination of different enzyme classes may not yield synergism during biomass hydrolysis, as the GH family allocation of the enzymes influences their behavior. As a result, it is important to understand which GH family combinations are compatible to gain knowledge on how to efficiently depolymerize biomass into fermentable sugars. We evaluated GH10 (Xyn10D and XT6) and GH11 (XynA and Xyn2A) β-xylanase performance alone and in combination with various GH family α-l-arabinofuranosidases (GH43 AXH-d and GH51 Abf51A) and α-d-glucuronidases (GH4 Agu4B and GH67 AguA) during xylan depolymerization. No synergistic enhancement in reducing sugar, xylose and glucuronic acid released from beechwood xylan was observed when xylanases were supplemented with either one of the glucuronidases, except between Xyn2A and AguA (1.1-fold reducing sugar increase). However, overall sugar release was significantly improved (≥1.1-fold reducing sugar increase) when xylanases were supplemented with either one of the arabinofuranosidases during wheat arabinoxylan degradation. Synergism appeared to result from the xylanases liberating *xylo*-oligomers, which are the preferred substrates of the terminal arabinofuranosyl-substituent debranching enzyme, Abf51A, allowing the exolytic β-xylosidase, SXA, to have access to the generated unbranched *xylo*-oligomers. Here, it was shown that arabinofuranosidases are key enzymes in the efficient saccharification of hetero-xylan into xylose. This study demonstrated that consideration of GH family affiliations of the carbohydrate-active enzymes (CAZymes) used to formulate synergistic enzyme cocktails is crucial for achieving efficient biomass saccharification.

## 1. Introduction

Xylan-containing lignocellulosic biomass is a renewable, carbon-neutral energy source that has considerable potential as a feedstock for the large-scale production of second-generation liquid fuels [1]. Xylan is a hetero-polysaccharide, primarily made up of xylose residues that are connected via β-1,4-glycosidic bonds [2,3]. The backbone chain is substituted with different side chains, such as glucuronopyranosyl, 4-*O*-methyl-d-glucuronopyranosyl, α-l-arabinofuranosyl, acetyl, feruloyl and *p*-coumaroyl residues, to varying degrees [3,4,5].

Xylan is present in many different plant species; however, its abundance and composition may vary between plants [6,7]. Based on their composition, xylans are classified as arabinoxylans (AXs), glucuronoxylans (GXs) and arabinoglucuronoxylans (AGXs) or glucuronoarabinoxylans (GAXs). AXs and GAXs are the main non-starch hemicelluloses found in the cell walls of endosperms of Gramineae such as wheat, bamboo, rice, sorghum, sugarcane and ryegrass [8,9]. On the other hand, hardwood and softwood species generally have the majority of their xylan-type of hemicelluloses being GXs and AGXs, respectively [3,4].

In plants, xylan is closely associated with cellulose and lignin through the formation of covalent and non-covalent linkages, thereby providing a rigid and protective structure for the plant cell wall [10]. In as much as xylan-containing lignocellulosic biomass has potential as a feedstock for the production of second-generation fuels, the presence of xylan polysaccharides that cover and are intertwined with cellulose is a barrier to cellulosic ethanol production, as the presence of xylan results in the resistance of biomass to conversion into fermentable sugars [11].

To accomplish efficient and complete xylan hydrolysis, the synergistic action of various xylanolytic enzymes is required [12,13]. The two key xylanolytic enzymes required for this degradation process are *endo*-1,4-β-d-xylanases (EC 3.2.1.8) and 1,4-β-d-xylosidases (EC 3.2.1.37) [3,12]. *Endo*-1,4-β-d-xylanases are responsible for cleaving internal β-1,4-glycosidic bonds in the xylan backbone into shorter oligosaccharides, whereas 1,4-β-d-xylosidases initiate their mode of action from the non-reducing end of the chain, hydrolyzing xylooligosaccharides to d-xylose [12]. Xylanases have been classified into glycoside hydrolase families GH5, 8, 10, 11, 30, 43, 62 and 98 in the CAZy database, with GH10 and 11 being the best-characterized families [14,15,16,17]. According to the CAZy database, xylosidases are presently found in glycoside hydrolase families GH3, 30, 39, 43, 52, 54, 116 and 120 [18].

In addition, the activities of accessory enzymes are required to liberate the substituents from the xylan backbone [5,19]. These enzymes are α-l-arabinofuranosidases (Abfs) and α-d-glucuronidases (EC 3.2.1.131) [20]. Abfs remove L-arabinose (L-Araf) residues from the backbone [2]. Abfs belong to families GH2, 3, 10, 43, 51, 54 and 62 in the CAZy database. Abfs that release L-Araf units from mono-substituted main-chain d-xylopyranosyl (d-Xylp) motifs are termed AXH-m, while those that release single L-Araf residues from double-substituted main-chain d-Xylp motifs are termed AXH-d. The α-d-glucuronidases (EC 3.2.1.131) are known to catalyze the hydrolysis of the α-1,2-linkages between xylose and d-glucuronic acid residues [2]. According to Nurizzo et al. [21], some GH4 and 67 glucuronidases prefer short glucuronic acid-substituted *xylo*-oligomers and are generally intracellular or membrane-associated, while others have specificity for polymeric glucuronic acid-containing polymeric xylans, these being consequently extracellularly produced enzymes. Generally, GH115 glucuronidases are more active on polymeric glucuronic acid-containing polymeric xylans [22,23], compared to GH4 and 67 glucuronidases [24,25].

As global interest in using xylan-containing lignocellulosic feedstocks for second-generation biofuel production increases, accompanying knowledge on how to efficiently depolymerize these feedstocks into fermentable sugars is required. Since enzymes classified with the same Enzyme Commission (EC) that are allotted to different glycoside hydrolase (GH) families can display different mechanisms of action and substrate specificities; for example, short xylooligosaccharides and soluble (branched) xylan-active GH10 xylanases versus insoluble xylan-active GH11 xylanase [3]. It is therefore expected that a combination of different enzyme classes may not yield synergism or synergize to comparable extents during biomass hydrolysis, as the GH family allocation of the enzymes influences their behavior. As a result, it is important to understand which GH family combinations are compatible to gain knowledge on how to efficiently depolymerize biomass into fermentable sugars. Research seeking to understand the synergism of various xylanases (GH10 and GH11) with other xylanolytic enzymes, Rex and debranching enzymes, such as feruloyl esterases, for efficient xylan degradation has been conducted by our research group. However, the intricate role of GH family allocations of debranching enzymes on their synergism with xylanases from various GH families during hetero-xylan degradation is not well established. The current study sought to unravel synergistic associations using GH10 and GH11 *endo*-xylanases from various organisms with various EC and GH family debranching enzymes, α-l-arabinofuranosidases (GH43 and GH51) on wheat flour arabinoxylan and α-d-glucuronidases (GH4 and GH67) on beechwood glucuronoxylan for the efficient degradation of these hemicellulosic substrates.

## 2. Results and Discussion

### 2.1. Substrate Specificity Determination

All the xylanolytic enzymes used in the study were assessed for their physicochemical properties (pH optima, temperature optima and thermal stabilities) before substrate specificity studies were conducted. The pH optima of the enzymes fell in a range of between 5.0 and 7.0, with SXA and Xyn10D exhibiting the lowest pH optimum of 5.0, while AguA exhibited the highest pH optimum of 7.0. Both α-arabinofuranosidases, Abf51A and AXH-d, and xylanases, Xyn2A and XT6, displayed pH optima of 6.0. The enzymes were found to have temperature optima around 40 °C. The two fungal GH11 xylanases, Xyn2A and XynA, and GH51 α-arabinofuranosidase, Abf51A, all displayed maximal activity at 50 °C (Table 1). The GH10 xylanase, Xyn10D, displayed a temperature optimum of 60 °C. The bacterial GH10 xylanase, XT6, and α-arabinofuranosidase, GH43 AXH-d, displayed maximal activity at 70 °C. The other xylanolytic enzymes, besides Xyn2A and XT6, displayed high thermal stability at 37 °C compared to 50 °C (data not shown). The enzymes were then tested for specific activities using different substrates at 37 °C and 50 mM sodium citrate buffer, pH 5.0, since these conditions assured fairly good and stable activity (the enzymes retained more than 80% relative activity) of all the enzymes assessed in this study (Table 1).

When evaluating the substrate specificities of the α-arabinofuranosidases, Abf51A displayed the highest arabinofuranosidase activity only on *p*NPA (161.9 U/mg), while AXH-d exhibited the highest activity on wheat flour arabinoxylan (30.9 U/mg), followed by *p*NPA (2.6 U/mg). Similar to *Bifidobacterium* sp. GH43 AXH-d, recombinant AXHd3 from B. adolescentis displayed the highest activity on wheat flour arabinoxylan (28.3 U/mg) followed by *p*NPA (0.1 U/mg) [26]. The α-glucuronidase, AguA, exhibited the highest glucuronidase activity on aldouronic acids (5.7 U/mg) followed by beechwood xylan (1.6 U/mg). Contrary to our findings, Nagy et al. [24] and Nurizzo et al. [21] noted that the enzyme does not release 4-*O*-methyl-glucuronic acid from glucuronoxylan. The enzyme, AguA, also displayed arabinofuranosidase cross activity on *p*NPA (2 U/mg). The α-glucuronidase, Agu4B on the other hand exhibited negligible activity on all the considered xylan-based substrates (Table 2). Similarly, a previous study showed that Agu4B was non-xylanolytic, as it has no detectable activity towards 4-*O*-methyl-d-glucuronoxylan and its fragment oligosaccharides [27]. The β-xylosidase, SXA, showed the highest activity (155.6 U/mg) on *p*NPX and some minor arabinofuranosidase activity (19.8 U/mg) on *p*NPA (Table 2). Jordan and Li [28] similarly reported that SXA was 10-fold more active on *p*NPX compared to *p*NPA. All the xylanases displayed the highest activity on wheat flour arabinoxylan, followed by beechwood xylan (Table 2). GH11 xylanases (XynA and Xyn2A) showed substantially higher hydrolytic activity on the xylan substrates, beechwood xylan and wheat flour arabinoxylan, than did GH10 xylanases (Xyn10D and XT6) (Table 2). Our previous studies showed similar results for the specific activities of most of the GH10 and GH11 enzymes [5,29].

### 2.2. Degradation Products of GH10 and 11 Xylanases on Xylans

Generally, GH10 xylanases are reported to attack the glycosidic linkage next to a single or double-substituted xylose toward the non-reducing end and require two unsubstituted xylose residues between branched residues, while GH11 xylanases are said to only hydrolyze glycosidic bonds where the two corresponding xylose moieties in subsites (−1) and (+1) are not branched [15,30]. Rosa et al. [31] have also proposed that the two GH family *endo*-xylanases (GH10 and 11) produce different types of oligosaccharides as hydrolysis products, as a result of differences in the cleavage specificities of the enzymes. GH10 xylanases generally release oligomers, in which the sugar at the non-reducing end is decorated with methyl-glucuronate, while those from family 11 liberate oligomers with internal methyl-glucuronate substituents during glucuronoxylan degradation [31].

Hydrolysis products as a result of the activities seen for XynA, Xyn2A, Xyn10D and XT6 on xylans (beechwood and wheat flour) were evaluated. Figure 1a,b indicate that these enzymes are true *endo*-xylanases as they mainly release XOS with a degree of polymerization of two and above. The GH11 family xylanase, Xyn2A, derived from *T. viride* also released small quantities of xylose during beechwood and wheat flour xylan degradation (Figure 1a,b); such a feature has been reported for other fungal xylanases, such as *Thermomyces lanuginosus* xylanase reported by Gomes et al. [32]. A homolog to the *T. viride* Xyn2A used in this study, XYNII from *T. reesei*, was shown to have five putative subsites: three (+1, +2, +3) at the reducing end and two at the nonreducing end (−1, −2) [33]. Furthermore, it was shown that xylooligosaccharides smaller than or equal to xylotetraose (X4) bound to XYNII in a manner in which the first (−2) or the last subsite (+3) is empty [33]; this implies that the hydrolysis of X4 would yield either X2 or X and X3.

Also, during beechwood xylan degradation, Xyn10D and XT6 additionally released an oligosaccharide, which migrated between xylotetraose and xylopentaose (Figure 1a)—we anticipate that this product is a xylooligosaccharides substituted with an α-glucuronic acid side chain (particularly aldopentaouronic acid). Xyn2A, on the other hand, appeared to only produce aldopentaouronic acid and no xylotetraose, while XynA exclusively produced xylotetraose (Figure 1a). Previously, aldotetraouronic acid has been shown to migrate between xylotriose and xylotetraose during TLC development [29,34]; we therefore hypothesize that aldopentaouronic acid is the compound migrating between xylotetraose and xylopentaose. Sydenham et al. [35] have demonstrated that GH10 xylanases generate high quantities of uronic acids compared to GH11 xylanases during hardwood (beech and birch) xylan degradation.

Generally, the hydrolysis products produced by xylanase using wheat flour (Figure 1b) were relatively different from those produced when beechwood xylan was used as a substrate (Figure 1a). Similarly, Sydenham et al. reported on this phenomenon when they evaluated hydrolysis products by two GH10 members, designated GtXyn10A and GtXyn10B, and two GH11 members, designated OpXyn11A and CcXyn11C, during beechwood, birchwood and wheat flour xylan degradation [35]. Similar to findings by Sydenham and others, family 10 xylanases produced identical hydrolysis product profiles when using the same substrate (Figure 1a,b, lanes 1 and 2), whereas the product profiles produced by the two GH11 xylanases differed from one another (Figure 1a,b, lanes 3 and 4) and the profiles produced by the two GH10 xylanases, particularly during beechwood xylan degradation (Figure 1a, lanes 1–4). Most of the auxiliary enzymes described so far mostly act only on substituted oligosaccharides produced by *endo*-xylanases, and only a few have been identified with activity on the polymeric xylan; therefore, knowing the hydrolysis products of the xylanases will shed light in unraveling the mechanism of their synergistic interaction with these debranching auxiliary enzymes.

### 2.3. Synergy Studies

Enzyme synergism is a phenomenon that occurs when interactions between two or more hydrolytic enzymes produce a total effect greater than the sum of the effects of the individual enzymes [10,36]. In this study, enzyme synergism using xylanases from GH families 10 and 11 during their interaction with α-l-arabinofuranosidases from GH families 43 and 51, as well as α-d-glucuronidases from GH families 4 and 67, was investigated during the degradation of wheat flour arabinoxylan and beechwood glucuronoxylan, respectively.

#### 2.3.1. Synergism during Beechwood Glucuronoxylan Degradation

*O*-acetyl-(4-*O*-methylglucurono)-xylans are the main hemicelluloses present in traditional industrial hardwoods, with their content varying between 15 and 30% (dry weight basis) [3]. Complete degradation of these xylans requires the synergistic action of a consortium of glycoside hydrolase enzymes, including *endo*-xylanase, β-xylosidase and α-glucuronidase. It appears, in highly decorated xylans, as if the presence of α-glucuronidase would assist by removing substituents that would have hindered the xylanase from cleaving the xylan backbone [31]. The action of the xylanase is then required to produce short, decorated oligomers, which are more attractive substrates for α-glucuronidase compared to decorated xylan polymers. It is proposed that these steps occur simultaneously and are interdependent of eacFh other during xylan depolymerization [37].

Combinations of Xyn2A and AguA at 93.75:6.25% and 75:25% protein dosage, respectively, exhibited synergistic cooperation and led to a significant (*p* < 0.05) improvement in reducing sugar release (approximately 3.5 mg/mL or 30.8% xylan saccharification) during beechwood xylan degradation compared to when Xyn2A was used alone at a 100% protein dosage (3.3 mg/mL reducing sugars or 29% xylan saccharification) (Figure 2b).

All other mixtures of the xylanolytic enzymes assessed showed no synergistic cooperation in reducing sugar, glucuronic acid and xylose release (Figure 2a,c,d and Figure 3). Moreover, Agu4B displayed no activity on beechwood xylan (Table 2), and as a result, it did not synergize with any of the xylanases (Figure 3). The 4-*O*-methyl-d-glucurono-d-xylan (MGX) represents more than 90% of the hemicellulose component in hardwoods with an average methyl-glucuronic acid:xylose ratio of 1:10 [38]. Based on the synergy studies, it was apparent that the hydrolysis of beechwood xylan with a combination of xylanases with glucuronidases did not improve saccharification yields compared to when xylanases were used alone. This shows that the low content of 4-*O*-methyl-d-glucuronic acid (10%) is negligible enough not to cause steric hindrance on the xylanases when they are acting on the xylan polymers. Glucuronoxylan degradation by the concerted action of xylanases and glucuronidases has been evaluated previously—from these studies, it has been demonstrated that the addition of the side-chain cleaving enzyme, glucuronidase, did not significantly improve the release of reducing sugars from these xylan substrates [31,34,39]. In contrast, the GH67 α-glucuronidase from *Dictyoglomus turgidum* (DtuAgu) was recently shown to significantly improve beechwood xylan conversion to xylose when evaluated using two mixtures of thermostable bacterial enzymes [40]. DtuAgu is distantly related to the enzyme used in this study, AguA from *G. stearothermophilus* T6; this may explain the differences in their substrate specificities and synergistic interactions with xylanases during glucuronoxylan degradation. Additionally, GH67 α-glucuronidases have been shown to have specificity for substrates in which 4-*O*-methylglucuronate is α-1,2-linked to the reducing terminal xylose in β-1,4-linked xylooligosaccharides [41]. We suspect that Xyn2A may be releasing this type of aldopentaouronic acid (i.e., UXXX) from beechwood xylan, while Xyn10D and XT6 may be releasing aldopentaouronic acid with internally linked 4-*O*-methylglucuronate, i.e., 2^3^-(4-*O*-methyl-α-d-glucuronyl)-xylotetraose (XUXX) or 2^2^-(4-*O*-methyl-α-d-glucuronyl)-xylotetraose (XXUX). It is also noteworthy to mention that the synergistic combinations of Xyn2A and AguA led to no improvement in glucuronic acid (approximately 0.1 mg/mL or 6% glucuronic acid release) and xylose release (approximately 1.5 mg/mL or 16.5% saccharification yield), and only reducing sugar release was significantly improved (Figure 2b). We presume that synergism resulted when AguA removed the decoration on 2^4^-(4-*O*-methyl-α-d-glucuronyl)-xylotetraose (UXXX) that would have hindered the action of Xyn2A from processing this oligosaccharide further into shorter oligosaccharides, i.e., X2 (more likely product) or X and X3 (unlikely since there was no xylose release improvement). On the other hand, the lack of synergism between AguA and the GH10 xylanases may result from AguA being unable to process the aldouronic acids in which the uronic acid is not linked to the non-reducing terminal xylopyranosyl residue but is internally linked to the oligosaccharide backbone [42]. Therefore, these xylanases would be unable to process this oligosaccharide into two X2 molecules. It is also noteworthy to mention that SXA could not assist hydrolysis by cleaving XUXX into X and UXX, which AgaA could act on, since GH43 xylosidases, such as the one from *G. stearothermophilus*, have narrow and shallow catalytic pockets which cannot accommodate branching, such as a uronic acid or arabinofuranoside, in xylose residues in either −1 or +1 subsites in the active site [43,44]. Moreover, SXA has been shown to accommodate sugar substitution to a xylooligosaccharide; if this occurs, two or three residues upstream the +1 site, more towards the reducing end of the xylooligosaccharide, which extends to the bulk solvent [45].

#### 2.3.2. Synergism during Wheat Flour Arabinoxylan Degradation

The addition of the GH51 α-l-arabinofuranosidase, Abf51A, particularly at a protein loading between 12.5 and 25%, with either one of the xylanases, led to significantly higher reducing sugar, xylose and arabinose release in comparison to when the xylanases were used alone (see Figure 4). Slightly lower synergism was observed between GH10 xylanases and Abf51A, with reducing sugar release up to 1.18-fold higher for Abf51A to XT6 (37.8% xylan saccharification) and up to 1.11-fold higher for Abf51A to Xyn10D (36.2% saccharification). However, slightly higher synergism was recorded with the GH11 xylanases, where an up to 1.24-fold reducing sugar improvement was recorded for XynA to Abf51A at 75:25% with 31% xylan saccharification and 1.32-fold reducing sugar improvement for Xyn2A and Abf51A at the same ratio with 34.8% saccharification.

Interestingly, the addition of GH43 α-l-arabinofuranosidase, AXH-d, with either one of the xylanases only led to significantly higher reducing sugar release (up to 1.25-fold higher for AXH-d to Xyn2A (33.5% saccharification), up to 1.16-fold higher for AXH-d to Xyn10D (35.7% saccharification), up to 1.5-fold higher for AXH-d to XT6 (37.9% saccharification) and up to 1.35-fold higher for AXH-d to XynA (21.7% saccharification)), while xylose and arabinose release were not improved (see Figure 5), except with Xyn10D to AXH-d at 93.75:6.25% and XynA to AXH-d at 87.5:12.5% protein loading, whereby xylose release was significantly improved up to 1.22-fold and 1.79-fold higher, respectively (see Figure 5a,c. Here, it appears that AXH-d exhibits similar specificity for intact arabinoxylan compared to arabinofuranosyl-branched *xylo*-oligomers generated by the action of xylanases on arabinoxylan—thus, no differences in arabinose release are displayed when AXH-d is used alone or in conjunction with the xylanases. Interestingly, xylose release was also not significantly improved when AXH-d was used in conjunction with the xylanases (Figure 5); we propose that AXH-d targets double-arabinofuranosyl substituted xylose residues in arabinoxylan, removing a single arabinose unit while the other one is left behind to impede the xylosidase, SXA, action. This phenomenon appears to occur with Abfs, which release single L-Araf residues from double-substituted main-chain D-Xylp motifs, as Sorensen et al. reported that the action of Abf II, a GH43 enzyme from *Humicola insolens*, did not expose additional xylan stretches for a xylosidase and xylanase to act on during arabinoxylan degradation, which in turn indicated that the Abf II catalyzed the release of arabinose from doubly substituted xylose residues [46]. On the other hand, Abf51A targets single arabinofuranosyl-substituted xylose residues in arabinoxylan, removing this single arabinose unit, as a result allowing the xylosidase, SXA, to be able to act on these regions—thus the improvement in xylose release (see Figure 4). However, it is noteworthy to mention that AXH-d releases over 20-fold more arabinose from arabinoxylan compared to Abf51A (Figure 4 and Figure 5). In the future, it would be interesting to evaluate how a combination of these two mechanistically different α-arabinofuranosidases, Abf51A and AXH-d, would interact with and influence the hydrolytic potential of backbone xylan cleaving enzymes xylanase and xylosidase. It is noteworthy to mention that GH10 xylanases appeared to have a significant influence on the release of monosaccharides xylose and arabinose by the xylosidase, SXA and arabinofuranosidases during arabinoxylan degradation. It seems that the GH10 xylanase products were the preferred substrates for these exolytic enzymes, xylosidase and arabinofuranosidases, in comparison to those released by GH11 xylanases.

It is noteworthy to mention that the current study only considered synergism between *endo*-xylanases and debranching enzymes, arabinofuranosidases and glucuronidases, with a minimum fixed amount of xylosidase. This may explain the low saccharification yields achieved during the saccharification of the hetero-xylans (between 30 and 40%). For efficient hetero-xylan degradation and improved saccharification yields, we propose the following: (1) fine-tuning the xylosidase ratio, (2) swapping the glucuronidases with the xylan polymer specific GH115 glucuronidase, (3) the addition of an *exo*-xylanase (Rex) and accessory enzymes, such as glucuronoyl esterases, feruloyl esterases, acetyl xylan esterases and xylan-specific lytic polysaccharide monooxygenases, (4) the exploration of synergism using the GH5 arabinoxylanases and GH30 glucuronoxylanases and (5) fine-tuning both the protein loading and hydrolysis time.

## 3. Materials and Methods

### 3.1. Materials and Enzymes

Beechwood-(Xyl:GlcAOMe:Other sugars = 80.8:11.4:7.8%) and wheat flour arabinoxylan (Ara:Xyl = 38:62%), xylobiose, xylotriose, xylotetraose, xylopentaose, xylohexaose, aldouronic acids mixture (aldotriouronic, aldotetraouronic and aldopentaouronic acids; (2:2:1)), α-glucuronidase (AguA, GH67) from *Geobacillus stearothermophilus* T6, a β-xylosidase from *Selenomonas ruminantium* (SXA, GH43), a *Bifidobacterium* sp. α-arabinofuranosidase (AXH-d,GH43), a *Clostridium thermocellum* arabinofuranosidase (Abf51A,GH51), a *Cellvibrio japonicas* β-xylanase (Xyn10D, GH10) and a *Geobacillus stearothermophilus* T6 β-xylanase (XT6, GH10) were all purchased from Megazyme™ (Bray, Ireland). A *Thermotoga maritima* α-glucuronidase (Agu4B, GH4) was purchased from NZYTech Genes & Enzymes (Lisbon, Portugal). The *p*-nitrophenyl substrates (*p*-nitrophenyl-α-d-arabinofuranoside, *p*NPA and *p*-nitrophenyl-β-d-xylopyranoside, *p*NPX), xylose and xylanase from *Trichoderma viride* (Xyn2A, GH11) were all purchased from Sigma Aldrich (St. Louis, MO, USA). A GH11 β-xylanase, XynA, from *Thermomyces lanuginosus* VAPS-24 (Gene Accession Number: KU366607) was produced at the Enzyme Technology and Protein Bioinformatics Laboratory, Department of Microbiology, Maharshi Dayanand University, Rohtak 124001, Haryana, India and purified in-house, as described previously [17,47].

### 3.2. Protein Determination

Bradford’s method was used to determine the quantity of protein content in the supplied enzymes [48]. Bovine serum albumin was used as a suitable protein standard.

### 3.3. Substrate Specificity Determination

Xylanase activity was determined using the polymeric substrates, beechwood flour arabinoxylan and wheat flour arabinoxylan. The xylanase activities were assayed in 400 µL reaction mixtures (1% *w*/*v* final substrate concentration) containing 300 µL of 1.33% (*w*/*v*) substrate dissolved in 50 mM citrate buffer (pH 5.0) and 100 µL of enzyme solution. The reaction was conducted at 37 °C for 30 min followed by centrifugation at 16,060× *g* for 5 min. The reducing sugar released was monitored with the DNS (dinitrosalicylic acid) method, as described by Miller [49]. About 300 µL of DNS was added to 150 µL of each supernatant sample. This was later followed by boiling on a heating block at 100 °C for 5 min, after which it was cooled on ice for 5 min. A single standard curve for the DNS assay was prepared using xylose as a suitable standard. Enzyme activity was measured in units (U), where 1 unit was defined as the amount of enzyme releasing 1 μmol of reducing sugar per min.

Alpha-glucuronidase activity assay reactions were set up the same way as the xylanase activity assays; however, a mixture of aldouronic acids was used as a substrate and the release of glucuronic acid was measured enzymatically using a D-galacturonic/D-glucuronic acid assay kit (K-URONIC, Megazyme). Enzyme activity was measured in units (U), where 1 unit was defined as the amount of enzyme releasing 1 μmol of glucuronic acid per min.

The β-xylosidase activity was performed using 2 mM *p*NPX. The assay reaction mixture was set up in a ratio of 1:9 with 50 µL of appropriately diluted enzyme: 450 µL of 2.25 mM *p*NPX in 50 mM sodium citrate buffer (pH 5.0) (2 mM *p*NPX final concentration). The reaction was conducted at 37 °C for 15 min and terminated by the addition of 500 µL of 2 M sodium carbonate. For α-arabinofuranosidase activity determination, the same assay as mentioned for β-xylosidase activity was used; however, *p*NPA was used as a substrate. The released *p*-nitrophenyl product was measured at 405 nm. Enzyme activity was measured in units (U), where 1 unit was defined as the amount of enzyme releasing 1 μmol of *p*-nitrophenol per min.

### 3.4. Determination of Degradation Products of Xylanases

A 400-microliter reaction mixture containing 0.25 mg of a xylanolytic enzyme (XynA, Xyn2A, Xyn10D and XT6) per g of biomass and 1% (*w*/*v*) of xylan (beechwood xylan and wheat flour arabinoxylan) in 50 mM citrate buffer (pH 5.0) was incubated at 37 °C for 6 h. No β-xylosidase, SXA, was supplemented in these reactions, as we wanted to assess the hydrolysis products profiles of the individual xylanases. After enzymatic hydrolysis, the samples were inactivated by heating for 5 min at 100 °C. About 5 μL of the samples were applied to Silica Gel 60 F254 TLC plates (Merck, Darmstadt, Germany). The plates were then developed twice with 1-butanol:acetic acid:water (2:1:1, *v*/*v*/*v*). The sugars developed on the plates were finally visualized by heating at 110 °C for 10 min after soaking the plates in Molisch’s Reagent (0.3% (*w*/*v*) α-naphthol dissolved in sulfuric acid: methanol solution (5: 95, *v*/*v*)).

### 3.5. Synergy Studies

In the enzyme synergy studies, enzyme mixtures were used in various combinations of either a GH10 or GH11 xylanase with a debranching enzyme to formulate hetero-synergistic binary enzyme mixtures. For beechwood xylan degradation synergy studies, Agu4A and AguA were used as the debranching enzymes in the hetero-synergism studies. For wheat flour arabinoxylan degradation synergy studies, Abf51A and AXH-d were used as the debranching enzymes in the hetero-synergism studies. The enzyme protein mass ratios considered were 100:0%, 93.75:6.25%, 87.5:12.5%, 75:25% and 0:100% xylanase to debranching enzyme. All the enzyme mixtures contained xylanolytic enzymes loaded at a total protein loading of 0.25 mg of enzyme per g of xylan, with *S. ruminantium* β-xylosidase, SXA, added at 10% of this loading (0.025 mg per g of xylan) to produce xylose from the hydrolysates. The experiments were carried out with 1% (*w/v*) xylan (beechwood xylan or wheat flour arabinoxylan) in 50 mM sodium citrate buffer (pH 5.0) in a 400 µL total reaction volume at 37 °C, mixing at 25 rpm for up to 24 h. Analysis for reducing sugar release was conducted according to the methods described in Section 2.3.

Quantitative analysis of arabinose, glucuronic acid and xylose concentration in the synergism assay hydrolysates were then performed enzymatically (K-ARGA, K-URONIC and K-XYLOSE, Megazyme) with the incubation time (20 min, twice the manufacturer’s recommended time) as this gave more reproducible results with the microtiter plate format. The saccharification yield of xylan was calculated using the following equation:hydrolysis of xylan % = ((mg/mL of sugar × 0.88)/(mg of sugar per 10 mg/mL xylan)) × 100(1)
whereby sugar can represent a monosaccharide (such as arabinose, glucuronic acid and xylose) or reducing sugar.

### 3.6. Data Analysis

One-way analysis of variance (ANOVA) was used to analyze the activity of the enzymes. The evaluation was conducted to elucidate significant increases exhibited by the enzyme combinations for reducing sugar and monosaccharide release (compared to that released by either one of the enzymes at 100% protein enzyme loading). All pairwise comparison procedures were based on a 95% confidence level (*p* < 0.05) and were conducted with the Data analysis feature in Microsoft^®^ Excel. All reactions were conducted in quadruplicate (*n* = 4), and error bars represent standard deviations.

## 4. Conclusions

The current study contributes to a body of knowledge generated from a series of investigations that sought to understand the synergism of various xylanases (GH10 and GH11) with other xylanolytic enzymes, Rex and debranching enzymes, such as feruloyl esterases, for efficient xylan degradation, which have been conducted by our research group [5,29,50]. The current study showed that α-glucuronidases do not significantly improve glucuronoxylan degradation by xylanases, while α-arabinofuranosidases synergized significantly with xylanases during wheat flour arabinoxylan degradation. The GH family of the α-arabinofuranosidase greatly influenced the degree of cooperability with the xylanases. The study demonstrated that it was possible to empirically design a “minimal” enzyme cocktail for efficient xylan hydrolysis by blending various GH family enzymes active on different types of bonds in hetero-xylan to achieve effective saccharification.

## Figures and Tables

**Figure 1 molecules-26-06770-f001:**
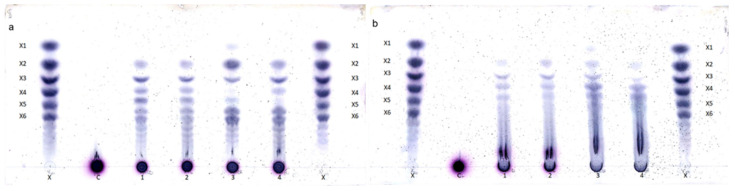
TLC analysis of hydrolysis of (**a**) beechwood xylan and (**b**) wheat flour arabinoxylan by (1) Xyn10D, (2) XT6, (3) Xyn2A, and (4) XynA for 6 h at 37 °C. A mixture of xylooligosaccharides (xylose [X1] to xylohexaose [X6]) [X] was used as standards. Un-hydrolyzed xylans (C) were also included as negative controls.

**Figure 2 molecules-26-06770-f002:**
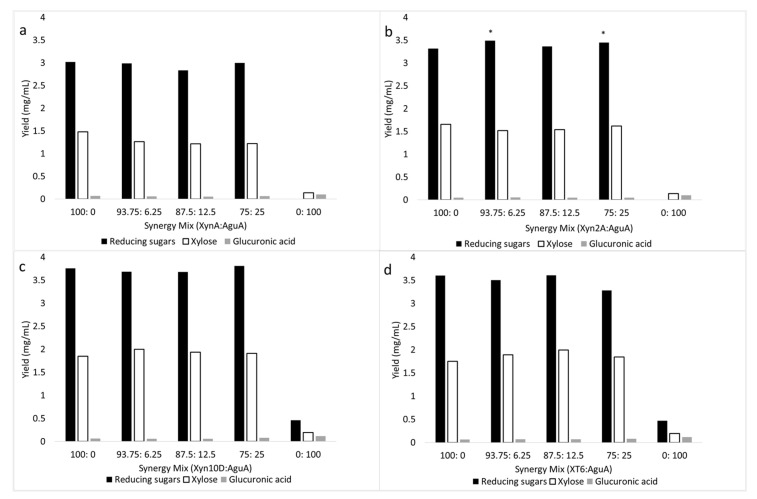
Comparative hydrolysis of beechwood xylan by (**a**) XynA and AguA, (**b**) Xyn2A and AguA, (**c**) Xyn10D and AguA, and (**d**) XT6 and AguA (lone or mixed at various ratios). ANOVA analysis for improvement of hydrolysis for reducing sugar and xylose release by the enzyme combinations compared to lone enzyme protein loading, keys: * (*p* value < 0.05). Values are represented as mean values ± SD (*n* = 4).

**Figure 3 molecules-26-06770-f003:**
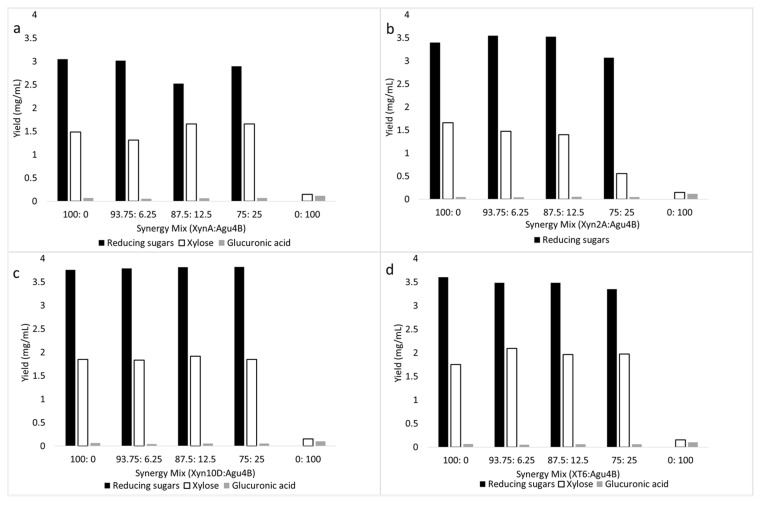
Comparative hydrolysis of beechwood xylan by (**a**) XynA and Agu4B, (**b**) Xyn2A and Agu4B, (**c**) Xyn10D and Agu4B, and (**d**) XT6 and Agu4B (lone or mixed at various ratios). ANOVA analysis for improvement of hydrolysis for reducing sugar and xylose release by the enzyme combinations compared to lone enzyme protein loading. Values are represented as mean values ± SD (*n* = 4).

**Figure 4 molecules-26-06770-f004:**
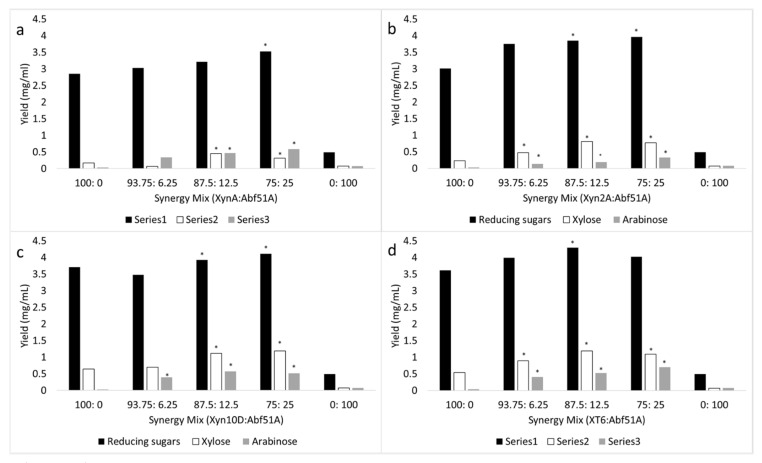
Comparative hydrolysis of wheat flour arabinoxylan by (**a**) XynA and Abf51A, (**b**) Xyn2A and Abf51A, (**c**) Xyn10D and Abf51A, and (**d**) XT6 and Abf51A (lone or mixed at various ratios). ANOVA analysis for improvement of hydrolysis for reducing sugar and xylose release by the enzyme combinations compared to lone enzyme protein loading, keys: * (*p* value < 0.05). Values are represented as mean values ± SD (*n* = 4).

**Figure 5 molecules-26-06770-f005:**
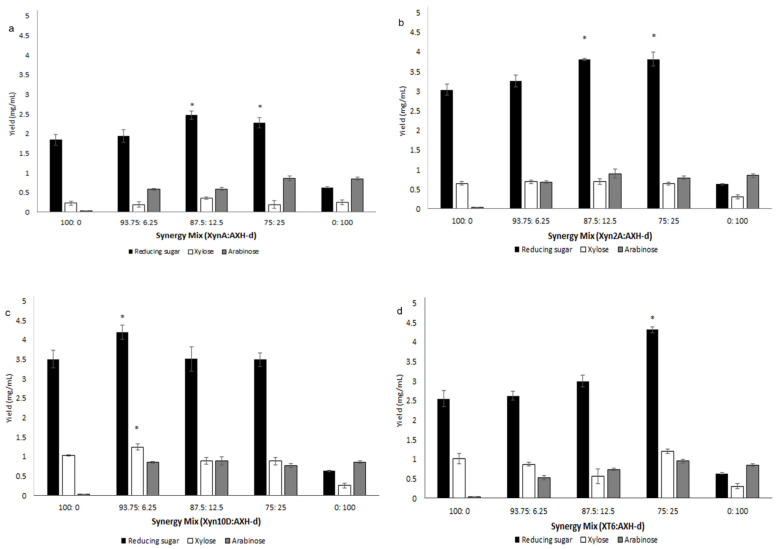
Comparative hydrolysis of wheat flour arabinoxylan by (**a**) XynA and AXH-d, (**b**) Xyn2A and AXH-d, (**c**) Xyn10D and AXH-d, and (**d**) XT6 and AXH-d (lone or mixed at various ratios). ANOVA analysis for improvement of hydrolysis for reducing sugar and xylose release by the enzyme combinations compared to lone enzyme protein loading, keys: * (*p* value < 0.05). Values are represented as mean values ± SD (*n* = 4).

**Table 1 molecules-26-06770-t001:** Properties of the studied xylanolytic enzymes. Where Nd = not determined.

Property	Xylanolytic Enzyme Assessed								
	Abf51A	AXH-d	AguA	Agu4B	SXA	XynA	Xyn2A	Xyn10D	XT6
**GH family**	51	43	67	4	43	11	11	10	10
**Mass (kDa)**	58.5	59.4	93.2	Nd	61.9	20.9	25.1	39	43.8
**pH optimum**	5.5	6.0	7.0	7.5	5.0	7.0	6.0	5.0	6.0
**Temperature optimum**	50	70	70	40	40	50	50	60	70

**Table 2 molecules-26-06770-t002:** Specific activities (U/mg protein) of the various xylanolytic enzymes on various xylan substrates. Where Nd = not determined.

Substrate	Xylanolytic Enzyme Assessed								
	Abf51A	AXH-d	AguA	Agu4B	SXA	XynA	Xyn2A	Xyn10D	XT6
** *p* ** **NPA**	161.9	2.6	2	Nd	19.8	Nd	2.1	Nd	2.1
** *p* ** **NPX**	0.4	0.3	0.2	Nd	155.6	Nd	0.4	Nd	0.4
**Aldouronic acids**	Nd	Nd	5.7	0.68	Nd	Nd	Nd	Nd	Nd
**Beechwood xylan**	0	0.23	1.6	0.96	3.9	91	258.4	30.8	141.8
**Wheat flour xylan**	0	30.9	0.9	0	6.6	370.6	502.1	133.1	273.6

## Data Availability

The data that support the findings of this study are available from the corresponding author, upon reasonable request.

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
