# Peer review of "Unraveling Synergism between Various GH Family Xylanases and Debranching Enzymes during Hetero-Xylan Degradation"

_molecules, 2021, doi:10.3390/molecules26226770_

Round 1
Reviewer 1 Report
See the attached file.

Author Response
The manuscript is an attempt to find optimum conditions for enzymatic saccharification of plant xylans by a combination of an enzyme cleaving the xylan main chain and one debranching glycosidase. The study with glucuronoxylan substrate could afford valuable data if endoxylanases would be used with GH115 α-glucuronidase that liberates uronic acid from internal xylopyranosyl residues. Unfortunately, GH10 and GH11 xylanases were used in combination with α-glucuronidases which do attack neither the polymeric substrate nor aldouronic acids liberated with the two xylanases. Therefore, the lack of synergism is not surprising and this is the reason why this part of the manuscript should be deleted.
>>Response to reviewer
Respectfully, we hold the view that the sharing these null and negative synergism results is vital for scientific progress in the field of biomass conversion, and as a result, we have decided to retain this section in the manuscript.
Indeed, the reviewer is correct that, perhaps higher synergistic effects would be observed if a GH115 glucuronidase was used with the two GH family allotted xylanases.
Possible ways to improve the saccharification of hetero-xylans by CAZymes has been proposed, see section 2.3.2:
“It is noteworthy to mention that, the current study only considered synergism between endo-xylanases and debranching enzymes, arabinofuranosidases and glucuronidases, with minimal fixed amount of xylosidase. This may explain the low saccharification yields achieved during the saccharification of the hetero-xylans (between 30 and 40%). For efficient hetero-xylan degradation and improved saccharification yields, we propose the following: (1) fine-tuning the xylosidase ratio, (2) swapping the glucuronidases with the xylan polymer specific GH115 glucuronidase, (3) addition of an exo-xylanase (Rex) and accessory enzymes, such as glucuronoyl esterases, feruloyl esterases, acetyl xylan esterases and xylan specific lytic polysaccharide monooxygenases, and (4) fine-tuning both the protein loading and hydrolysis time.”
Surprising is the extremely high yield of free xylose since no xylose liberation is seen on TLCs in Fig. 1. This contradiction is raised here despite difference in duration of incubations (6 h vs 24 h).
>>Response to reviewer
Indeed, no xylose is release in the TLC studies, as no β-xylosidase, SXA, was supplemented in these reactions, as we wanted to assess the hydrolysis products profiles of the individual xylanases, see section 3.4 in Materials and Methods. While for the synergy studies, the enzyme mixtures contained xylanolytic enzymes loaded at a total protein loading of 0.25 mg of enzyme per g of xylan, with S. ruminantium β-xylosidase, SXA, added at 10% of this loading (0.025 mg per g of xylan) to produce xylose from the hydrolysates, see section 3.5 in Materials and Methods.
The authors also demonstrate in the manuscript their insufficient knowledge of the mode of action of endoxylanases, particularly the nature of major aldouronic acids generated by Gh10 and GH11 xylanases. They did not explain the abbreviations of aldouronic acids used in the manuscript. E.g., does the formula UXXX correspond to xylotetraose substituted with uronic acid at the non-reducing end? If this so the oligosaccharide should be called aldopentaouronic acid, because it contains five aldoses one of which contains the carboxyl. The names of abbreviated compounds are not correct and some of the mentioned oligosaccharides represent just very minor products which may originate by cleavage of the
main chain close to its terminals. The information on major acidic products of xylanases generated from glucuronoxylan has been widely confirmed.
>>Response to the reviewer
The symbols of the acidic xylooligosaccharides referred to in the manuscript have been defined as 2³-(4-O-methyl-α-D-glucuronyl)-xylotetraose (XUXX), 22-(4-O-methyl-α-D-glucuronyl)-xylotetraose (XXUX) and 24-(4-O-methyl-α-D-glucuronyl)-xylotetraose (UXXX).
The part of the manuscript describing the effect of α-L-arabinofuranosidases looks better and contains an interesting observation that the addition of the GH43 enzyme, which liberates 3-O-linked Ara from doubly arabinosylated units leads to higher degree of arabinoxylan hydrolysis. A more detail study in this direction could generate valuable and exceptionally interesting results.
I my opinion the manuscript is not suitable for publication.
In general, we appreciate the comments of the reviewer because they enhanced our manuscript. We have addressed some of the comments. Please see the detailed responses to reviewers comments in the attached document.
Kind regards,
Mpho Mafa
Reviewer 2 Report
This manuscript discusses a straightforward, somehwat archival, series of studies to evaluate how combining glycosyl hydrolases from the GH10 and GH11 families might impact the enzyme-catalyzed degradation of various types of xylan. The studies are appropriate and the data included in the manuscript and SI support the authors’ assertions and conclusions. In addition, sufficient details are provided for other workers to reproduce the work. Although the standard of English is generally good, some grammatical editing is needed prior to any publication. Less positively, there are a number of points on which I am not clear. For example, the introduction is hard to follow for the non-expert, and certainly does not define the key problem that is being addressed nor explain why the results in this paper have anything other than archival significance. Although a lot of information is presented, I remain unclear as to why the present study is needed given what is already known about converting xylan into useful sugars. Does the work outlined in this paper represent a more efficient conversion or allow the breakdown of previously intractable xylans? What is the innovation here except for using a different combination of enzymes. Why did the authors choose the enzymes used in this study or except that they would be synergistic? It is all a bit of a mystery. This is especially the case given that two of these authors have already published a paper (Enzyme & Microbial Technology (2019), in which they report almost identical work demonstrating that GH10 and GH11 family members exhibit synergy in catalyzing the breakdown of xylan backbones. So what is new here?
I could, however, support publication in Molecules if major revisions are made that address these points and the comments given below.
Abstract: I have no idea why the first sentence is needed. It could be deleted without any loss of information. The definition of CAZymes also escapes me. Indeed, the importance of the study is never clearly state in the abstract.
I was surprised to see that some enzymes of bacterial origin had temperature optima of 70 oC but that the bacteria were not named. What are the Tm values for these enzymes? My perusal of the literature suggests that these are from the termite hindgut and it may be that they are archaeal enzymes. Can the authors comment on the stability of these enzymes and their host organisms in more detail?
Figures 2 & 3: The synergistic effect is reported for a 24 h incubation. Does the effect exhibit a time dependence? What is the % xylan that is hydrolyzed in these experiments, i.e. is 100% conversion observed at 24 h? If 100% conversion cannot be achieved (perhaps because product is inhibiting the enzymes) then when does the mixture reach equilibrium? I also wonder whether the extent of synergy depends on temperature. Was this investigated? Regarding this point, why was 37 oC chosen for these experiments when some of the combinations used enzymes with temperature optima of 50 oC (and above)?
Author Response
This manuscript discusses a straightforward, somewhat archival, series of studies to evaluate how combining glycosyl hydrolases from the GH10 and GH11 families might impact the enzyme-catalyzed degradation of various types of xylan. The studies are appropriate and the data included in the manuscript and SI support the authors’ assertions and conclusions. In addition, sufficient details are provided for other workers to reproduce the work. Although the standard of English is generally good, some grammatical editing is needed prior to any publication.
Grammatical editing where necessary has been conducted as suggested by the reviewer.
Less positively, there are a number of points on which I am not clear. For example, the introduction is hard to follow for the non-expert, and certainly does not define the key problem that is being addressed nor explain why the results in this paper have anything other than archival significance. Although a lot of information is presented, I remain unclear as to why the present study is needed given what is already known about converting xylan into useful sugars. Does the work outlined in this paper represent a more efficient conversion or allow the breakdown of previously intractable xylans? What is the innovation here except for using a different combination of enzymes? Why did the authors choose the enzymes used in this study or except that they would be synergistic? It is all a bit of a mystery. This is especially the case given that two of these authors have already published a paper (Enzyme & Microbial Technology (2019)), in which they report almost identical work demonstrating that GH10 and GH11 family members exhibit synergy in catalyzing the breakdown of xylan backbones. So, what is new here?
The published paper cited by the reviewer explored synergism between xylan backbone cleaving enzymes, endo-xylanases and exo-xylanases, also called Rexs, while the current study explored debranching enzyme to endo-xylanase synergism.
The significance of the study has been quoted in the introduction as suggested by the reviewer, and reads as follows:
“Since enzymes classified with the same Enzyme Commission (EC) that are allotted to different glycoside hydrolase (GH) families can display different mechanisms of action and substrate specificities, for example, short xylooligosaccharides and soluble (branched) xylan active GH10 xylanases versus insoluble xylan active GH11 xylanase [3]. It is, therefore, expected that a combination of different enzyme classes may not yield synergism or synergize to comparable extents during biomass hydrolysis, as the GH family allocation of the enzymes influences their behavior. As a result, it is important to understand which GH family combinations are compatible to gain knowledge on how to efficiently depolymerize biomass into fermentable sugars. Research seeking to understand the synergism of various xylanases (GH10 and GH11) with other xylanolytic enzymes, Rex and debranching enzymes, such as feruloyl esterases, for efficient xylan degradation has been conducted by our research group. However, the intricate role of GH family allocations of debranching enzymes on their synergism with xylanases from various GH families during hetero-xylan degradation is not well established.”
I could, however, support publication in Molecules if major revisions are made that address these points and the comments given below.
Abstract: I have no idea why the first sentence is needed. It could be deleted without any loss of information.
The first sentence has been retained, however, two additional sentences have been added to clearly convey the significance of the study in the abstract, and they read as follows:
“Enzymes classified with the same Enzyme Commission (EC) that are allotted in different glycoside hydrolase (GH) families can display different mechanisms of action and substrate specificities. Therefore, combination of different enzyme classes may not yield synergism during biomass hydrolysis, as the GH family allocation of the enzymes influences their behavior. As a result, it is important to understand which GH family combinations are compatible to gain knowledge on how to efficiently depolymerise biomass into fermentable sugars.”
The definition of CAZymes also escapes me. Indeed, the importance of the study is never clearly state in the abstract.
The term “CAZymes” has been defined as carbohydrate-active enzymes.
I was surprised to see that some enzymes of bacterial origin had temperature optima of 70oC but that the bacteria were not named. What are the Tm values for these enzymes? My perusal of the literature suggests that these are from the termite hindgut and it may be that they are archaeal enzymes. Can the authors comment on the stability of these enzymes and their host organisms in more detail?
The names of the organisms which the enzymes were derived from are detailed in section 3.1 of the Materials and Methods section.
Not much was dwelled upon the biochemical and physico-chemical characteristics of the enzymes, as they are well defined, available commercially, and such data has been published before.
Figures 2 & 3: The synergistic effect is reported for a 24 h incubation. Does the effect exhibit a time dependence?
Indeed, synergism exhibits time dependence, our previous work has reported on this, see Malgas et al. (2017). Time dependence of enzyme synergism during the degradation of model and natural lignocellulosic substrates. Enzyme and Microbial Technology 103, 1-11.
What is the % xylan that is hydrolyzed in these experiments, i.e. is 100% conversion observed at 24 h? If 100% conversion cannot be achieved (perhaps because product is inhibiting the enzymes) then when does the mixture reach equilibrium?
The saccharification yields by the synergistic combinations are provided in the manuscript as suggested by the reviewer, and most are around 30% (see sections 2.3.1 and 2.3.2 for details).
Many factors influence the yield from not reaching 100% - such as low enzyme loading used, hydrolysis period, and also accessory enzymes for complete biomass deconstruction being required, i.e. carbohydrate esterases (glucuronyl-, feruloyl- and acetyl-xylan-esterases) and lytic polysaccharide monooxygenases.
I also wonder whether the extent of synergy depends on temperature. Was this investigated? Regarding this point, why was 37 oC chosen for these experiments when some of the combinations used enzymes with temperature optima of 50 oC (and above)?
Indeed, synergy is affected by temperature, as operating at temperatures which abolishes the activity of one of the enzymes in the combination led to no synergism being observed. Hence, a moderate temperature, which assured the activity of all enzymes, was selected.
We thank the reviewer for their positive and constructive comments. The comment has improved our manuscript, and please find our responses to the comments in the attached documents.
Many thanks,
Mpho Mafa
Round 2
Reviewer 1 Report
The authors corrected the manuscript, however, not in the way a reviewer would expect. The authors demonstrate that they are still not familiar with papers describing the types of aldouronic acids generated by GH10 and GH11 xylanases from glucuronoxylan. Just to remind them – GH10 xylanases produce mainly aldotetraouronic acid UXX (according to abbreviations used in the manuscript), which carries the uronic acid at the non-reducing end xylopyranosyl residue of xylotriose. UXX should serve as a substrate of GH67 α-glucuronidase. That means that the lack of synergy serves as evidence that the α-glucuronidase was inactive. The same is true for the reaction mixtures containing GH11 with α-glucuronidase in the case the GH43 β-xylosidase was present (the presence of β-xylosidase in reaction mixtures should be indicated in the figure legends). The β-xylosidase should shorten the main aldopentauronic acid XUXX, produced by GH11 xylanases to aldotetraouronic acid UXX which should serve as the substrate of GH67 α-glucuronidase. These are the facts which should lead to elimination of the whole part of the manuscript dedicated to glucuronoxylan hydrolysis. In case the authors would insist on interpretation of their data, they should first report the unusual performance of GH10 and GH11 xylanases which would be a more important novelty for the field than the manuscript data.
Author Response
Please find our responses to the valuable comments of the reviewe-1 in the attached document. Indeed, the comments have strengthened our work. Even though it seems like this is back and forth process, we value the comments of the review because they improved the paper and help clarify our argument.
Best regards
Mpho
